# A Golgi Apparatus-Targeting, Naphthalimide-Based Fluorescent Molecular Probe for the Selective Sensing of Formaldehyde

**DOI:** 10.3390/molecules26164980

**Published:** 2021-08-17

**Authors:** Maxine Mambo Fortibui, Wanyoung Lim, Sohyun Lee, Sungsu Park, Jinheung Kim

**Affiliations:** 1Department of Chemistry and Nano Science, Ewha Womans University, Seoul 120-750, Korea; maxinemambo@gmail.com (M.M.F.); lshyun127@ewhain.net (S.L.); 2Department of Global Biomedical Engineering, Sungkyunkwan University, Suwon 16419, Korea; wanyoung22@gmail.com; 3Department School of Mechanical Engineering, Sungkyunkwan University, Suwon 16419, Korea

**Keywords:** fluorescent probe, formaldehyde detection, phenylsulfonamide, condensation reaction, two-photon excitation

## Abstract

Formaldehyde (FA) is a colorless, flammable, foul-smelling chemical used in building materials and in the production of numerous household chemical goods. Herein, a fluorescent chemosensor for FA is designed and prepared using a selective organ-targeting probe containing naphthalimide as a fluorophore and hydrazine as a FA-binding site. The amine group of the hydrazine reacts with FA to form a double bond and this condensation reaction is accompanied by a shift in the absorption band of the probe from 438 nm to 443 nm upon the addition of FA. Further, the addition of FA is shown to enhance the emission band at 532 nm relative to the very weak fluorescent emission of the probe itself. Moreover, a high specificity is demonstrated towards FA over other competing analytes such as the calcium ion (Ca^2+^), magnesium ion (Mg^2+^), acetaldehyde, benzaldehyde, salicylaldehyde, glucose, glutathione, sodium sulfide (Na_2_S), sodium hydrosulfide (NaHS), hydrogen peroxide (H_2_O_2_), and the *tert*-butylhydroperoxide radical. A typical two-photon dye incorporated into the probe provides intense fluorescence upon excitation at 800 nm, thus demonstrating potential application as a two-photon fluorescent probe for FA sensing. Furthermore, the probe is shown to exhibit a fast response time for the sensing of FA at room temperature and to facilitate intense fluorescence imaging of breast cancer cells upon exposure to FA, thus demonstrating its potential application for the monitoring of FA in living cells. Moreover, the presence of the phenylsulfonamide group allows the probe to visualize dynamic changes in the targeted Golgi apparatus. Hence, the as-designed probe is expected to open up new possibilities for unique interactions with organ-specific biological molecules with potential application in early cancer cell diagnosis.

## 1. Introduction

Formaldehyde (FA) is both the simplest aldehyde and a highly reactive carbonyl species that is well known for its applications in food, textiles, and wood processing [1,2,3]. Indeed, FA exists at low levels in most living organisms and can be found in natural foods such as fruits, vegetables, meats, seafood, and dairy products [4]. However, FA has been recognized as the third largest indoor chemical pollutant, and exposure to FA may induce severe central nervous system damage, cancer, and even death [5,6,7]. Due to its toxicity, the United States Environmental Protection Agency has established a maximum recommended daily dose of 0.2 mg/kg per day for FA [8].

Biological FA is produced at levels of 0.2–0.4 mM that are maintained endogenously via enzymatic processes [7]. Whereas cell proliferation can be promoted, and memory formation mediated, at healthy FA levels, cognitive impairments, neurodegeneration, and memory loss have been reported at elevated concentrations due to potent protein and DNA cross-linking mechanisms.

Recently, several methods such as high performance liquid chromatography (HPLC), mass spectrometry, and colorimetry have been used to detect FA [9,10,11]. However, these traditional techniques have several disadvantages, including low sensitivity, complex operative procedures, long reaction times, and damage to living cells. In recent years, fluorescent probes have been recognized as powerful analytical tools for the detection of various ions and biomolecules in living systems with their high selectivity, sensitivity, fast response time, and in-situ biological applications [12,13,14].

Optical imaging is recognized as a supportive noninvasive technique for the study of living systems. Biological studies have suggested that the generation of FA is associated with many organelles, including lysosomes, the endoplasmic reticulum, and mitochondria [15,16,17,18]. However, the majority of fluorescent probes are limited in their applications in living tissues due to their use of one-photon excitation with short wavelengths, along with other shortcomings. By contrast, two-photon microscopy has the ability to penetrate deep tissue and could provide improved three-dimensional imaging with long-wavelength excitation [15,19].

Nevertheless, there remains a lack of information on the generation and distribution of FA in subcellular parts. Hence, the development of an organelle-targeted FA fluorescent probe would contribute to the enhanced understanding of FA activity in specific organelles. The Golgi body within a living cell plays a key role for transporting and secreting some important glycoproteins and lipids. In this respect, two interesting fluorescent probes for FA were recently reported [15,19]. However, these probes are not very specific as they can be taken up by almost all sub-organelles and exhibit Pearson coefficients of 0.60 ± 0.05 and 0.426 in the Golgi apparatus. Hence, there is still a strong need for the development of an efficient probe that is highly specific for a particular sub-organelle and the development of FA-targeted fluorescent probes with a high specificity towards the Golgi body is of significant importance for revealing the biological processes associated with various diseases. Meanwhile, H_2_O_2_- and H_2_S-targeted fluorescent probes incorporated with the phenylsulfonamide moiety have been reported to stain the Golgi body, showing high Pearson coefficients over 0.92 [20,21].

Herein, a new fluorescent probe is reported for staining Golgi apparatus and detecting FA with high specificity. The probe is designed to contain a phenylsulfonamide moiety as a Golgi-targeting group, 1,8-naphthalimide as a chromophore, and hydrazine as a reaction site for FA. The probe is shown to exhibit no significant fluorescence via a photoinduced internal electron transfer (PET) pathway. However, after the introduction of FA, the PET pathway is suppressed via the formation of a hydrazone group and thus a significant turn-on signal can be achieved. In addition, the as-developed probe is shown to afford two-photon emission upon the addition of FA with a good staining of the Golgi apparatus in live cells.

## 2. Results and Discussion

### 2.1. Design and Synthesis of Probe EW2

The condensation of an aldehyde group with fluorescent probes containing an amine or hydrazine functional group was used to develop chemosensors for the detection of formaldehyde (FA). As shown in Scheme 1, the probe consisted of a hydrazine-containing naphthalimide fluorophore with an FA-binding site along with a phenylsulfonamide group for targeting the Golgi apparatus [15,18,20,21].

### 2.2. Optical Properties of Probe EW2

The absorption spectra of probe **EW2** in the presence of various concentrations of FA (0–100 μM) are presented in Figure 1. Thus, in the absence of FA, the probe exhibits strong absorption bands at 260 and 442 nm in 10 mM phosphate-buffered saline (PBS, pH 7.4) containing 1 % dimethyl sulfoxide. With the addition of FA, the absorption band is seen to increase in absorbance, and a minor blue-shift to 440 nm is observed. The additional data in Appendix A also reveals a progressive increase in the blue-shift with increasing concentrations of FA.

The fluorescence emission measurements in Figure 1b indicate that the probe itself is almost completely non-fluorescent (Φ_F_ = 0.039). However, when FA is introduced to probe EW2, a strong fluorescence enhancement is observed (Φ_F_ = 0.415) at 546 nm and gradually increases with increasing concentrations of FA (0–100 µM). A good linear relationship (*R^2^* = 0.9842) is also established between the emission intensity and the concentration of FA (Appendix A). The detection limit was calculated to be 0.35 µM using the 3σ method. The additional data in Appendix A also demonstrate that probe EW2 exhibits two-photon emission properties towards FA. Moreover, the kinetic response of probe EW2 to FA was also studied, and the fluorescence enhancement was recorded in the presence of 5 and 10 equiv. of FA. In each case, the emission reached a maximum value within three minutes. As shown in Appendix A, the rate constant of the probe in the presence of 10 equiv. FA was determined to be *k* = 0.55 min^−1^.

To examine the emission properties of probe EW2 in biological conditions, the influence of pH upon the fluorescence response was studied in the absence and presence of FA. In the absence of FA, the probe exhibited a negligible change in fluorescence intensity as the pH varied from pH 4.0 to 10.0 (Appendix A). However, an enhancement in the fluorescence intensity with increasing pH was observed in the presence of FA, thus suggesting that FA can be detected by the probe under physiological pH conditions. These results indicate that the probe has application potential for the real-time imaging of FA in living systems.

### 2.3. Selectivity of EW2 towards FA

To evaluate the selectivity of probe EW2 towards FA, the fluorescence response towards a range of competing biological species was also examined, as shown in Figure 2. Here, the probe EW2 is seen to be highly selective towards FA, with a distinct fluorescence enhancement at 547 nm, over other analytes, such as Br^−^, Cl^−^, F^−^, I^−^, NO_2_^−^, NO_3_^−^, SCN^−^, OAc^−^, HSO_3_^−^, OH^−^, t*-*BuO^−^, OCl^−^, O_2_, H_2_O_2_, HS^−^, cysteine (Cys), homocysteine (Hcys), glutathione (GSH), 4-hydroxybenzaldehyde, 4-nitrobenzaldeyde, acetaldehyde, terephthaldehyde, and glyoxal. This result clearly indicates the excellent selectivity of the probe EW2 towards FA over other competing species for possible biological applications.

In addition, an experiment was performed in which FA was inhibited by the addition of NaHSO_3_. As shown in Appendix A, a progressive decrease in fluorescence was observed when FA (50 μM) was pretreated with increasing concentrations (50, 100, and 200 μM) of NaHSO_3_, followed by incubation with 5 μM EW2. These results demonstrate that the emission enhancement is diminished by the reaction of FA with NaHSO_3_.

### 2.4. Proposed Sensing Mechanism

To further verify the reaction product derived from the probe after the addition of FA, the final solution was characterized by electrospray ionization (ESI) mass spectrometry. As shown in Appendix A, a new peak was observed at *m*/*z* = 395.58 upon the addition of FA. The species observed is formulated as [3 + H^+^]^+^, thus confirming the formation of the hydrazone product via the reaction with FA (Scheme 2).

### 2.5. Detection of Formaldehyde in Food Samples

To evaluate the performance of the probe EW2 in the detection of FA in food, various samples (dried shiitake mushroom and onions) were prepared as explained in the Experimental Section. The food samples were then treated with EW2 (5 μM) and the corresponding emission intensities were recorded. As indicated in Appendix A, the mushroom and onion extracts responded to the probe with intensity enhancements, thus revealing the presence of certain amounts of FA.

To determine the amount of FA in the food extracts, the test samples were spiked with various amounts of FA (5, 10, and 15 µM), and the fluorescence intensity of all these samples were obtained. As shown in Table 1, the level of FA in the dried shiitake mushroom was found to be 5.2 µM, while that in onions was 4.8 µM. FA in these samples can be detected with probe EW2, and the recovery shows over 80 %. These results demonstrate that the EW2 probe can be used for the quantitative detection of FA in food.

### 2.6. Cell Imaging of the Probe

The probe EW2 was used to image FA in MCF7 cells. Negative control cells were treated with either FA alone or EW2 alone, with both controls affording no fluorescence (Figure 3). However, when the MCF7 cells were pre-treated with 50 μM FA for 30 min followed by 5 μM EW2 for an additional 40 min, strong fluorescence signals were observed. These results indicate that the EW2 probe can be used to detect FA in living cells. Moreover, the MCF7 cells treated with various concentrations of FA (5, 50, and 500 μM) revealed a significant dose-dependent turn-on fluorescence. These results confirm the ability of the probe EW2 to assay the FA quantitatively in living cells.

### 2.7. Cytotoxicity Studies

The feasibility of using probe EW2 to image FA in living cells was investigated. The cytotoxicity of EW2 was first evaluated via the MTT assay, as shown in Figure 4. In this assay, the MCF7 cells were treated with various concentrations (0–30 μM) of EW2. The results indicate that probe EW2 possesses low toxicity towards living cells and is suitable for imaging living organisms under the selected conditions. In addition, cell viability was verified using Hoechst 33342 staining. As shown in Figure 3d, a bright-field image was acquired and overlaid with a fluorescence image of the cells stained with Hoechst 33342. The result clearly reveals the presence of intact and viable nuclei after the treatment of probe EW2.

### 2.8. Inhibition Studies of the Probe

The inhibiting effect of NaHSO_3_ on FA was then investigated. In this set of experiments, the cells were treated with NaHSO_3_ as a negative control, and their fluorescence in the green channel was examined. As shown in Appendix A, this negative control afforded no fluorescence. However, when the cells were pre-treated with 500 µM FA for 20 min followed by 5 µM of probe EW2 for 40 min, an intense green fluorescence was observed. By contrast, no fluorescence was observed for the cells that were pre-treated with 500 µM FA and NaHSO_3_ for 20 min followed by 5 µM of EW2 for 40 min. These results are also consistent with the fluorescence results presented in Appendix A. Thus, the scavenging effect of NaHSO_3_ upon FA was demonstrated, confirming that the probe can be used to monitor exogenous FA in living cells.

### 2.9. Colocalization Experiment

Colocalization experiments were performed to investigate the specificity of the probe EW2 towards the Golgi apparatus afforded by the introduction of a phenyl sulfonamide group. In this experiment, the MCF7 cells were labelled with EW2 for 30 min and followed by staining of the Golgi apparatus with a BODIPY TR Ceramide for 15 min. As shown in Figure 5, the resulting green fluorescence signal from the EW2 exhibits a good degree of overlap with the red fluorescence of the BODIPY TR Ceramide. In addition, the correlation plot of the fluorescence intensity of the two channels in Figure 5 reveals a Pearson’s coefficient of 0.806, demonstrating that the EW2 specifically targets the Golgi apparatus. The imaging for the detection of FA in the Golgi apparatus suggests that it is the organelle where the toxic substance FA is detoxified. Moreover, the probe EW2 exhibits much higher specificity towards the Golgi apparatus than other FA probes [15,22].

### 2.10. Imaging Endogenous FA in Cells

Endogenous FA production is triggered by endoplasmic reticulum stress, which is known to cause various diseases, including neurodegeneration, atherosclerosis, type-2 diabetes, liver disease, and cancer [23]. To induce FA production, the cells were treated with thapsigargin (TG) prior to being stained with EW2 [23]. As the concentration of TG increased, the green fluorescence intensity became stronger (Figure 6), demonstrating that EW2 is also highly useful for imaging FA produced in cells.

## 3. Materials and Methods

### 3.1. Materials and Instrumentation

Unless indicated otherwise, all reagents were purchased from Aldrich and used without further purification. Distilled water was purified with a Milli-Q purification system and used throughout all experiments. The ^1^H-NMR and ^13^C-NMR spectra were recorded on a Bruker AVANCE III 300 MHz NMR spectrometer using DMSO-*d_6_* as the solvent and tetramethylsilane (TMS) as the internal reference. The absorption spectra were recorded on a Perkin-Elmer model Lambda 2S UV/Vis spectrometer. The emission spectra were recorded on a Shimadzu RF6000 Spectro-Fluorophotometer with a 1 cm standard quartz cell.

### 3.2. Synthesis of Compound 1

Compound **1** was synthesized according to the reported procedure [21]. 4-Bromo-1,8-napthalic anhydride (2.0 mmol) and 4-(2-aminoethyl)-pyridine (2.0 mmol) were dissolved in acetic acid (20 mL) and the reaction mixture was refluxed for 8 h. The mixture was then cooled to room temperature and the precipitate was filtered and dried to afford compound **1** as a grey solid (68%). ^1^H-NMR (300 MHz, DMSO-d_6_, ppm): δ 8.63 (dd, *J* = 8.0 Hz, 2H), 8.39 (d, *J* = 7.9 Hz, 1H), 8.26 (d, *J* = 7.9 Hz, 1H), 8.04 (d, *J* = 8.5, 7.4 Hz, 1H), 7.99 (dd, *J* = 8.4 Hz, 2H), 7.63(dd, *J* = 8.8 Hz, 2H), 7.53(s, 2H). ^13^C-NMR (300 MHz, DMSO-d_6_): δ 163.64, 163.58, 133.18, 132.06, 131.85, 131.42, 130.37, 129.66, 129.50, 129.36, 129.31, 129.20, 128.77, 123.81 (Appendix A). ESI mass, *m*/*z* = 432.17 for [**1** + H^+^]^+^, (calculated 432.26) (Appendix A).

### 3.3. Synthesis of Probe EW2

Compound **1** (0.2 mmol) and a solution of 80% hydrazine hydrate (0.5 mL) in 2 mL ethanol were refluxed for 8 h, then cooled to room temperature. The resulting precipitate was filtered and purified by chromatography on silica gel (DCM/MeOH=30:1) to afford the probe EW2 as an orange solid (70%). ^1^H-NMR (300 MHz, DMSO-d_6_, ppm): 8.61 (dd, *J* = 8.0 Hz, 2H), 8.38 (d, *J* = 7.9 Hz, 1H), 8.29 (d, *J* = 7.9 Hz, 1H), 8.07 (d, *J* = 8.5, 7.4 Hz, 1H), 7.99 (dd, *J* = 8.4 Hz, 2H), 7.63(dd, *J* = 8.8 Hz, 2H), 7.53(s, 2H). ^13^C-NMR (300 MHz, DMSO-d_6_): 163.43, 163.41, 149.90, 133.53, 132.17, 131.33, 131.18, 130.66, 130.59, 128.94, 128.15, 124.31, 122.79, 121.91 (Appendix A). ESI mass: found at *m*/*z* = 383.75 for [EW2 + H^+^]^+^, (calculated 383.39) (Appendix A).

### 3.4. Absorption and Fluorescent Titration Measurements

A stock solution of probe EW2 (1 mM) was prepared in DMSO. Formaldehyde and other analytes were prepared in distilled water. For a typical optical study, a solution of EW2 was prepared in phosphate-buffered saline (pH 7.4)/DMSO solution (99/1, *v*/*v*). The absorption and fluorescent spectra were recorded upon addition of the analyte of interest. For all the fluorescence measurements, the excitation wavelength was 440 nm and the slit width 5 nm.

### 3.5. Preparation of Formaldehyde Source from Dried Shiitake Mushrooms

The stems of purchased dried shiitake mushrooms were removed and the remaining material was cut into small pieces (2 g) and stored in a centrifuge tube. After the addition of distilled water (25 mL), the tube was sealed and placed into an ultrasonic bath for extraction at 40 °C for 1 h. The sample was then centrifuged at 6000 rpm for 10 min, filtered, and the filtrate stored for subsequent use.

### 3.6. Detection of Formaldehyde Source from Dried Shiitake Mushrooms

Probe EW2 stock solution (1 mM) was diluted in PBS buffer (10 mM, pH 7.4) and DMSO. The filtrate of shiitake mushroom (100 μL) was then added to adjust the total volume of the solution to 3 mL, which thus contained 10 μM EW2. The fluorescence spectra were recorded with an excitation of 440 nm.

### 3.7. Preparation of Onion Extract and Detection of Formaldehyde

Onion juice was prepared by homogenizing 500 g of peeled yellow onion bulb with 500 mL of deionized water in a kitchen blender. The onion juice was adjusted to the required pH (3.5–6.5) using 1 M acetic acid and incubated at 50 °C. The onion extract was then filtered and centrifuged to remove the pulp before further use.

### 3.8. Cell Imaging

The breast cancer cell line (MCF7) was grown in Dulbecco’s Modified Eagle’s Medium (DMEM) (Thermo Fisher Scientific, Seoul, Korea) containing 10% fetal bovine serum (FBS) (HyClone Laboratories, Erie, PA, USA), 100 units/mL penicillin (Life Technologies, Carlsbad, CA, USA), and 100 μg/mL streptomycin (Life Technologies) in a 5% CO_2_ incubator at 37 °C. They were seeded in a 35 mm confocal dish for 24 h. Probe EW2 was freshly prepared as a 1 mM stock in DMSO and its concentration was adjusted with cell medium prior to use. The cells were first incubated with 5 μM EW2 at 37 °C for 40 min, then washed three times with PBS and incubated with formaldehyde (FA) at various concentrations (5 μM–500 μM) in cell medium at 37 °C for 30 min. Prior to the imaging, the cells were washed three times with PBS again. The cells treated with 500 μM FA were stained with 5 μg/mL Hoechst-33342 (Thermo Fisher Scientific, Seoul, Korea) for 5 min. Cell images were obtained using a fluorescent microscope (Deltavision^®^, GE Healthcare, Notre Dame, IN, USA) with an FITC (Fluorescent isothiocyanate) filter set (λ_ex_ = 440 nm, λ_em_= 500–550 nm).

To visualize endogenous FA, cells were first treated with TG (15 μM–45 μM) for 30 min to produce FA and then incubated with EW2 (5 μM) for 40 min. Cell images were obtained by a K1-Fluo confocal microscope (Nanoscope Systems, Inc., Daejeon, Korea) with either an FITC (fluorescent isothiocyanate) filter set (λ_ex_ = 440 nm, λ_em_= 500–550 nm) or a TRITC (tetramethylrhodamine) filter set (λ_ex_ = 561 nm, λ_em_ = 570–620 nm).

## 4. Conclusions

The design and synthesis of a new fluorescent probe for the reaction-based sensing of FA was described herein. The probe contained fluorophore and hydrazine moieties and was shown to be highly selective towards FA over other biologically relevant analytes. In addition, the two-photon fluorescent properties of the probe were observed for the sensing of FA. The probe exhibited favorable properties such as a high turn-on fluorescence, high sensitivity, and a low detection limit. The probe could also be used to detect FA in commercially available products such as shiitake mushrooms and onions. In biological application, the probe gave satisfactory results for monitoring changes in both exogenous and endogenous FA levels of cancer cells. Moreover, the probe was able to visualize dynamic changes in the Golgi apparatus due to cancer cell apoptosis with high specificity. Thus, the as-developed probe has the potential to serve as an effective tool for the monitoring of FA in relation to the possible early diagnosis of endoplasmic reticulum stress in cells.

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
