# Peer review of "A Golgi Apparatus-Targeting, Naphthalimide-Based Fluorescent Molecular Probe for the Selective Sensing of Formaldehyde"

_molecules, 2021, doi:10.3390/molecules26164980_

Round 1

Reviewer 1 Report

This manuscript introduced a newly designed FA detecting fluorescent probe. Comparing with other previously developed FA probes and FA detecting methods, the EW2 reported in this study could selectively target the Golgi body and therefore has the advantage to provide information of FA levels in this specific organelle. The authors evaluated the optical properties and selectivity of EW2 in this manuscript and successfully used it to detect and quantify FA in food samples. For its application in cells, EW2 showed a co-localization with Golgi apparatus and the ability to detect pre-incubated FA with minimal cytotoxicity. It is also applicable for two-photon imaging that extends its potential in living tissues.

However, with the advantage of Golgi body specific targeting of EW2, the authors did not clearly state the significance or essentiality of monitoring FA levels in this one organelle. Except for experiments on food samples, all other tests of the probe EW2 were carried with exogenously incubated FA. It is not shown whether it could be applied to detect endogenous FA produced under physiological or pathological conditions.

Besides, when examining the optical properties of EW2, it is a little confusing that some experiments were done with single-photon imaging while some were two-photon imaging. The fluorescence emission measurement was done with 0-100μM FA, but there is not a clear scale for values of the corresponding fluorescent intensity or a concentration-intensity curve. Thus, it is hard to get a general view of EW2’s performance in response to different concentrations of FA. Meanwhile, according to the physiological or pathological FA concentration dynamic range, it is doubtful whether EW2 would be suitable for in vivo application. In addition, in Figure S3, the scale of emission intensity was provided while the corresponding FA concentrations were not consistent in the two subplots.

There are also several minor comments:

  1. In Figure 3, are the cells still alive with intact plasma membrane after FA treatment? And in Figure 3b, some fluorescent signals seem not to overlap with the cell.
  2. In Figure 5 a-c, the contrast is not suitable and the background signals are too noisy. The morphology of the Golgi body cannot be seen clearly as well as the co-localization.
  3. In figure S4, the legend labelled that 10 equiv FA was applied for obtaining the rate constant k=0.55 min-1, while in the article, it was described as 5 equiv FA.

Author Response

This manuscript introduced a newly designed FA detecting fluorescent probe. Comparing with other previously developed FA probes and FA detecting methods, the EW2 reported in this study could selectively target the Golgi body and therefore has the advantage to provide information of FA levels in this specific organelle. The authors evaluated the optical properties and selectivity of EW2 in this manuscript and successfully used it to detect and quantify FA in food samples. For its application in cells, EW2 showed a co-localization with Golgi apparatus and the ability to detect pre-incubated FA with minimal cytotoxicity. It is also applicable for two-photon imaging that extends its potential in living tissues.

However, with the advantage of Golgi body specific targeting of EW2, the authors did not clearly state the significance or essentiality of monitoring FA levels in this one organelle. Except for experiments on food samples, all other tests of the probe EW2 were carried with exogenously incubated FA. It is not shown whether it could be applied to detect endogenous FA produced under physiological or pathological conditions.

>> Thank you for the valuable advice. We discussed the signficance of monitoring of FA levels in Golgi appartus in the revised manuscript.

“The imaging for the detection of FA in Golgi apparatus suggests that it is the organelle where the toxic substance FA is detoxified.”

As the reviewer suggested, we have carried out cell imaging experiments using  thapsigargin (TG). The images of endogenous FA are presented in Figure 6. For these results, one more paragraph of 2.10 is added in the revised manuscript.

2.10. Imaging endogenous FA in cells.

Endogenous FA production is triggered by endoplasmic reticulum stress, which is known to cause various diseases, including neurodegeneration, atherosclerosis, type-2 diabetes, liver disease, and cancer [23]. To induce FA production, the cells were treated with thapsigargin (TG) prior to being stained with EW2 [23]. As the concentration of TG increased, the green fluorescence intensity became stronger (Figure 6), demonstrating that EW2 is also highly useful for imaging FA produced in cells.

Furthermore The Golgi apparatus plays a very important role in stress response and also when formaldehyde is accumulated in cells it can cause stress mostly known as formaldehyde stress(FA stress) in such a situation by carrying out cell experiments we can better understand how the Golgi apparatus will regulate the metabolism of formaldehyde in the living system. 

>> Line 41-44 explains the reason why it is important to design probes for the detection of formaldehyde and rather than targeting just living cells, it is essential to be able to direct new probes to specific cell organelles.

In addition, metabolism of methylated amines, including the abundant endogenous metabolite methylamine, by semicarbazide-sensitive amine oxidases (SSAO) releases FA and given the fact that the Golgi apparatus is known for its role as a central station in cells, receiving secretory content from the endoplasmic reticulum (ER), packing proteins into membrane-bound vesicles and secreted to extracellular environment, it is there important to be able to monitor the released FA.

Besides, when examining the optical properties of EW2, it is a little confusing that some experiments were done with single-photon imaging while some were two-photon imaging. The fluorescence emission measurement was done with 0-100μM FA, but there is not a clear scale for values of the corresponding fluorescent intensity or a concentration-intensity curve. Thus, it is hard to get a general view of EW2’s performance in response to different concentrations of FA. Meanwhile, according to the physiological or pathological FA concentration dynamic range, it is doubtful whether EW2 would be suitable for in vivo application. In addition, in Figure S3, the scale of emission intensity was provided while the corresponding FA concentrations were not consistent in the two subplots.

>> From the spectroscopic analysis, probe EW2 showed ability to detect FA both in one photon and two photon wavelength as presented in Figure 1b, S2 and S3a and S3b. We have edited our plots to include the scale values. We have corrected the caption of Figure S3; it has just one subplot which is a direction representation of the concentration dependent reaction of probe EW2 towards FA. In Figure S3, the concentration range of FA is 0.01 – 0.1 mM, but we made a mistake in the caption. The mistake has been corrected in the revised ESI.

There are also several minor comments:

  1. In Figure 3, are the cells still alive with intact plasma membrane after FA treatment? And in Figure 3b, some fluorescent signals seem not to overlap with the cell.

>> Yes, they were still alive. We replace the images with the new ones in Figure 3b.

  1. In Figure 5 a-c, the contrast is not suitable and the background signals are too noisy. The morphology of the Golgi body cannot be seen clearly as well as the co-localization.

>> Co-location imaging experiments were carried out with BODIPY TR Ceramide tracker dye and EW2 after TG treatment. The images in Figure 5 were replaced to the new ones with lesser background signals.

  1. In figure S4, the legend labelled that 10 equiv FA was applied for obtaining the rate constant k=0.55 min-1, while in the article, it was described as 5 equiv FA.

>> We made a mistake in the text. The rate constant was determined with 10 equiv. We have corrected it in the revised manuscript.

Reviewer 2 Report

Line 91: Scheme 1 is incorrect. Compound 1 cannot be prepared from the reactants described in the scheme. The scheme does not correspond at all to the described procedure of preparation 1 in lines 226 to 235. In this section, the authors describe that the reactants were stirred for 8 hours. at 160 ° C. This is not possible under normal pressure. The boiling point of acetic acid is 118 ° C. The whole preparation procedure 1 and also scheme 1 is incorrect.

The preparation of compounds similar to compound 1 has been published many times. It is necessary for the authors to cite some of these basic works.

Lines 236-245: The synthesis of compounds similar to EW2 was also published many times and therefore it is necessary to cite some of these works as well.

It is not necessary to present the DMSO signal intensity in the 1H NMR spectra of compounds 1 and EW2 (Fig. S10 and Fig. S13). The authors thus achieved a decreasing in the predictive value (readability) of the signals of compounds 1 and EW2. It is necessary to adjust the Y-coordinate so that the signals of compounds 1 and EW2 (Fig. S10 and Fig. S13) were as high as the DMSO signals.

Lines 246 to 250: the excitation wavelength of the sample and slits must be reported when measuring fluorescence spectra and quantum yields of fluorescence.

Line 123: „2.3. Selectivity of EW2 towards FA “ It is very surprising that the hydrazone formed by the reaction of EW2 with acetaldehyde does not have a comparable fluorescent signal as the hydrazone formed by the reaction of EW2 with FA. The results described by the authors in „2.4. Proposed sensing mechanism - lines 142 to 147 are listed in several places. I would therefore welcome instead of this text the addition of the text justifying the cause of the very low fluorescence signal of hydrazone formed by the reaction of EW2 with acetaldehyde.

Author Response

Line 91: Scheme 1 is incorrect. Compound 1 cannot be prepared from the reactants described in the scheme. The scheme does not correspond at all to the described procedure of preparation 1 in lines 226 to 235. In this section, the authors describe that the reactants were stirred for 8 hours. at 160 ° C. This is not possible under normal pressure. The boiling point of acetic acid is 118 ° C. The whole preparation procedure 1 and also scheme 1 is incorrect.

>> Thanks for the reviewer’ comment. We missed the amine group and have corrected Scheme 1. We did have the same concern regarding the high boiling point of acetic acid. However, we understand that, synthetic methods can be modified based on the reactants and interestingly we followed a referenced published article which actually carried out the reaction at 160 ° C (Hui, W.; Zixu, H.; Yuyun, Y.; Jiao Z.; Wei, Z.; Wen, Z.; Ping, L.; Bo, T. Ratiometric fluorescence imaging of 374 Golgi H2O2 reveals a correlation between Golgi oxidative stress and hypertension. Chem. Sci. 2019, 10, 10876-10880).

The preparation of compounds similar to compound 1 has been published many times. It is necessary for the authors to cite some of these basic works.

>> We have included the references 21-22 as the representative articles for the synthesis of compound 1.

Lines 236-245: The synthesis of compounds similar to EW2 was also published many times and therefore it is necessary to cite some of these works as well.

>> We have included the references 15 and 18 as the representative articles similar to probe EW2.

It is not necessary to present the DMSO signal intensity in the 1H NMR spectra of compounds 1 and EW2 (Fig. S10 and Fig. S13). The authors thus achieved a decreasing in the predictive value (readability) of the signals of compounds 1 and EW2. It is necessary to adjust the Y-coordinate so that the signals of compounds 1 and EW2 (Fig. S10 and Fig. S13) were as high as the DMSO signals.

>> The expanded spectra of compounds 1 and EW2 have been included in Figure S10 and S13.

Lines 246 to 250: the excitation wavelength of the sample and slits must be reported when measuring fluorescence spectra and quantum yields of fluorescence.

>> We have included this information in section 3.4. For all the measurements, the excitation wavelength was 440 nm and the excitation/emission slit widths were 5 nm.

Line 123: „2.3. Selectivity of EW2 towards FA “ It is very surprising that the hydrazone formed by the reaction of EW2 with acetaldehyde does not have a comparable fluorescent signal as the hydrazone formed by the reaction of EW2 with FA. The results described by the authors in „2.4. Proposed sensing mechanism - lines 142 to 147 are listed in several places. I would therefore welcome instead of this text the addition of the text justifying the cause of the very low fluorescence signal of hydrazone formed by the reaction of EW2 with acetaldehyde.

>> Such low fluorescence of the products of other FA probes with acetaldehyde were also reported in references 15 and 18.  

Round 2

Reviewer 2 Report

The reactants in acetic acid cannot be heated to 160 ° C at an atmospheric pressure of about 760 mm Hg column. If the authors refluxed the reaction mixture, the temperature of the reaction mixture reached a maximum value of 118 ° C. (it is the boiling point of acetic acid). The authors would only reach a temperature of 160 ° C if the acetic acid and 4- (2-aminoethyl) -pyridine in the reactor were in the gaseous state. In this case, only 4-Bromo-1,8-napthalic anhydride alone would be heated to 160 ° C. If the reaction was carried out under such conditions, then it is necessary to describe the reactor in detail.

The temperature of the reaction mixture (4-Bromo-1,8-napthalic anhydride + acetic acid + 4- (2-aminoethyl) -pyridine) 160 ° C can be reached e.g. in an autoclave, at a pressure greater than 760mm Hg.

Author Response

The reactants in acetic acid cannot be heated to 160 ° C at an atmospheric pressure of about 760 mm Hg column. If the authors refluxed the reaction mixture, the temperature of the reaction mixture reached a maximum value of 118 ° C. (it is the boiling point of acetic acid). The authors would only reach a temperature of 160 ° C if the acetic acid and 4- (2-aminoethyl) -pyridine in the reactor were in the gaseous state. In this case, only 4-Bromo-1,8-napthalic anhydride alone would be heated to 160 ° C. If the reaction was carried out under such conditions, then it is necessary to describe the reactor in detail.

The temperature of the reaction mixture (4-Bromo-1,8-napthalic anhydride + acetic acid + 4- (2-aminoethyl) -pyridine) 160 ° C can be reached e.g. in an autoclave, at a pressure greater than 760mm Hg.

Response:

We did not use 4-(2-aminoethyl)-pyridine in our synthesis. We don’t know why it was included in my synthetic procedure and used sulfanilamide.

As you can see in the reference attached below, they used 160 0C.

Anyway, we revised the synthesis part of compound 1, according to the reviewer’s comment.